# Solution Combustion Synthesis of Ni/Al₂O₃ Catalyst for Methane Decomposition: Effect of Fuel

Pavel B. Kurmashov [1], Arina V. Ukhina [2], Anton Manakhov [3,*], Arkady V. Ishchenko [4], Evgenii A. Maksimovskii [5] and Alexander G. Bannov [1,*]

1 Department of Chemistry and Chemical Engineering, Novosibirsk State Technical University, 630073 Novosibirsk, Russia; kurmaschov@gmail.com
2 Institute of Chemistry of Solid State and Mechanochemistry, Siberian Branch, Russian Academy of Sciences, 630092 Novosibirsk, Russia
3 Research Laboratory of Inorganic Nanomaterials, National University of Science and Technology (MISIS), Leniskiy Prospect, 4, 119049 Moscow, Russia
4 Boreskov Institute of Catalysis, Siberian Branch, Russian Academy of Sciences, 630090 Novosibirsk, Russia; arcady.ishchenko@gmail.com
5 Institute of Inorganic Chemistry, Siberian Branch, Russian Academy of Sciences, 630090 Novosibirsk, Russia; eugene@niic.nsc.ru
* Correspondence: ant-manahov@ya.ru (A.M.); bannov.alexander@gmail.com (A.G.B.)

**Abstract:** The synthesis of a 90% Ni/Al₂O₃ catalyst via solution combustion synthesis with various fuels was studied in this work. Catalysts with a high content of the active component (i.e., nickel) were obtained as a result of the combustion of $Ni(NO_3)_2 \cdot 6H_2O$ and $Al(NO_3)_3 \cdot 9H_2O$ mixtures with fuel. The fuels, such as hexamethylenetetramine, glycine, urea, starch, citric acid, and oxalic acid, were investigated. The synthesis was carried out in a furnace, with the temperature being raised from room temperature to 450 °C at a rate of 1 °C per min. The paper evaluates the efficiency of fuels and their effect on the structure and properties of catalysts, as well as their catalytic activity. The catalyst was used for the synthesis of hydrogen and carbon nanofibers by methane decomposition at 1 bar and 550 °C. The catalysts were tested in a vertical flow reactor without preliminary reduction. The obtained samples of catalysts and carbon nanomaterials were studied by transmission electron microscopy, low-temperature nitrogen adsorption, and X-ray diffraction. The highest activity of the catalyst was obtained when citric acid was used as a fuel. The specific yields of hydrogen and carbon nanofibers were 17.1 mol/g_cat and 171.3 g/g_cat, respectively. Catalytic decomposition of methane led to the formation of cup-stacked carbon nanofibers.

**Keywords:** methane; carbon nanofibers; hydrogen; solution combustion; fuel; carbon nanomaterials

## 1. Introduction

The processing of C₁–C₄ hydrocarbons is currently a relevant and efficient technology for producing hydrogen from various sources (associated petroleum gas, natural gas, methane, industrial mixtures of hydrocarbons in the petrochemical industry, oil refining, etc.). One promising approach is the catalytic decomposition of the hydrocarbons in the mixture ($CH_4 \rightarrow C + H_2$, $\Delta H° = 74.8$ kJ/mol). The process under consideration can be carried out at a variety of pressures and temperatures, producing two main products: hydrogen and carbon. The process is usually carried out over nanoparticle catalysts. The latter represent the metals of the iron subgroup of the periodic table. The advantages of the technology are its simplicity and the absence of carbon dioxide emissions in the reaction products. A CO_x-free process is one of the main advantages of hydrogen production compared to steam reforming [1]. In addition to the formation of hydrogen, a by-product of the reaction is carbon nanofibrous material [2]. The disadvantage is the deactivation of the catalyst, which occurs due to the deposition of solid reaction products (carbon) on the

active sites of the catalyst. Sometimes, $CO_x$-free hydrogen can also be produced by the decomposition of ammonia [3].

The structure and physical or chemical properties of carbon nanomaterials depend on the synthesis parameters (temperature and pressure), the composition of the reaction gas mixture (methane, propane, a mixture of hydrocarbons, etc.), the catalyst preparation technique, and the composition of catalyst. The activity of a catalyst directly depends on its chemical nature and composition, so the main components of the catalyst for the process under consideration are the metals of the iron subgroup (for example, the metals Ni, Co, and Fe). In addition, alloying the catalyst with Cu or other metals in a small amount makes it possible to increase catalytic activity by an order of magnitude [4]. The metals are most often evenly distributed on the oxide substrate (for example, $Al_2O_3$ [5], $SiO_2$ [6], MgO [7], and $ZrO_2$ [8]). Sometimes, the concentration of the active component (i.e., metal) in the catalyst is high enough compared to content of substrate. This composition of the catalytic system leads to the formation of granulated carbon nanofibrous material, representing the granules consisting of densely packed carbon nanofibers (CNFs) [9,10]. In some articles, this material is called nanofibrous carbon [11–13]. Initially, the research was devoted to the investigation of CNFs' synthesis and properties [14]. Then, the study was directed to the co-production of hydrogen and CNFs [15]. Carbon nanomaterials (namely, carbon nanofibers and carbon nanotubes), produced using a CVD $CO_x$-free approach to methane decomposition, can be used in various fields of application, for example, catalysis [16,17], supercapacitors [18,19], adsorption [20], polymer composites [21–24], gas sensors [10,25,26], etc.

There are many synthesis techniques of catalysts for the co-production of hydrogen and carbon nanomaterials via the decomposition of $C_1$–$C_4$ hydrocarbons. For example, sol-gel synthesis [27], mechanical activation [28], solution combustion synthesis [29], co-precipitation [30], electroexplosive synthesis [31], etc. Solution combustion synthesis has many good points when it comes to making catalysts for breaking down methane. The advantage of solution combustion synthesis (SCS) of catalysts for hydrogen production is a one-step process of preparation [29,32]. A majority of methods for catalyst preparation contain a greater number of steps compared to SCS.

In [33], a combination of hexamethylenetetramine and citric acid (so-called "mixed fuel") was used for the synthesis of CoCu, CoNi, CoFe, and FeNi nanomaterials. The microwave-assisted method was used for the production of Co nanoparticles using various fuels (citric acid, glycine, urea, and hexamethylenetetramine) [34]. However, despite the amount of research devoted to the solution combustion synthesis of nanoparticles, the results of their application in the creation of catalysts for methane decomposition are poorly presented, especially when considering methane conversion, hydrogen yield, and carbon nanomaterials formed over these catalysts.

This work was devoted to the SCS of a $Ni/Al_2O_3$ catalyst for the production of carbon nanofibers and hydrogen in a vertical flow reactor at a relatively low temperature (550 °C) and atmospheric pressure. A comparison of different fuels (hexamethylenetetramine, glycine, urea, starch, citric acid, and oxalic acid) and their effects on the efficiency of hydrogen production was carried out. SCS was carried out in a furnace by means of heating at a constant rate (1 °C per min). The analysis of texture properties, crystallite size, and phase composition of catalysts synthesized by SCS was performed.

## 2. Materials and Methods

Catalysts with a high content of the active component (i.e., nickel), $90Ni-10Al_2O_3$ (wt.%), were obtained by SCS. Salts of metals $Ni(NO_3)_2 \cdot 6H_2O$ and $Al(NO_3)_3 \cdot 9H_2O$, weighing 13.3757 g and 2.2075 g, respectively, were dissolved in distilled water (100 mL) until a homogeneous solution was obtained. Then, a fuel (also called the reducing agent) weighing 1.5 g was introduced into the resulting two-component solution, which was then dissolved until a homogeneous three-component solution, $Ni(NO_3)_2$–$Al(NO_3)_3$–$H_2O$–fuel, was formed. Next, the catalyst precursor (mixture) was calcined according to a given program in a SNOL 6.7/1300 muffle furnace. The calcination was carried out using the

programmable heating approach. The heat treatment was carried out by heating the sample from room temperature to 723 K (450 °C) at a heating rate of 1 °C per min, as a result of which the three-component solution was gradually dehydrated until a concentrated gel-like paste was formed. In the area of the sample, a stable front of a combustion wave arises in the solution as a result of the redox reactions, with the subsequent formation of nanoparticles of NiO and $Al_2O_3$. The heating was performed in air. The catalysts synthesized after calcination were crushed to a particle size fraction of 100–200 μm and then tested in a laboratory vertical quartz reactor.

The process of catalyst preparation by SCS is shown in Figure 1.

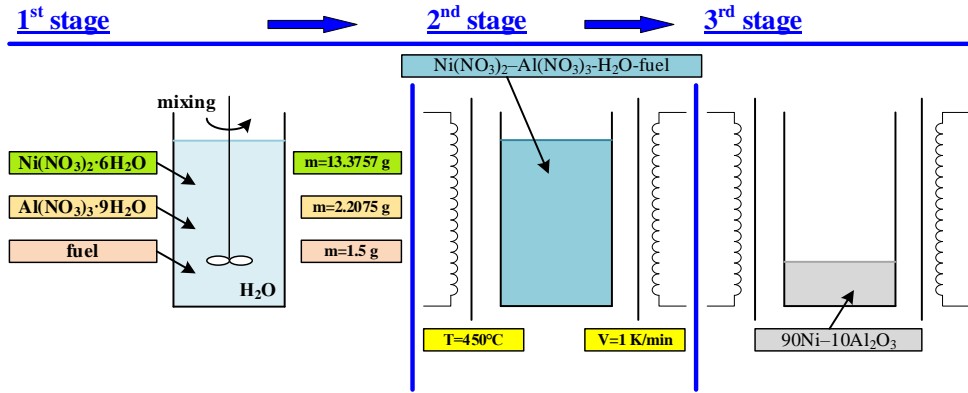

**Figure 1.** Preparation technique of Ni/$Al_2O_3$ catalyst using solution combustion synthesis.

Table 1 shows the experimental data on the synthesized catalyst samples, which were obtained using the following reducing agents: urotropin (hexamethylenetetramine, $C_6H_{12}N_4$) (HMTA), glycine ($C_2H_5NO_2$), urea ($CO(NH_2)_2$), starch ($C_6H_{10}O_5)_n$), citric acid ($C_6H_8O_7$), and oxalic acid ($C_2H_2O_4$).

**Table 1.** Summary data on the synthesized catalyst samples.

| Sample | Fuel Excess (φ) | Parameters of Catalyst Preparation in Furnace | | Results of Synthesis (450 °C, 1 atm) | | |
| --- | --- | --- | --- | --- | --- | --- |
| | | Temperature of Calcination, K | Heating Rate V, K/min | Total Time of Reaction t, h | Specific Yield of Hydrogen ($Y_{H2}$), mol/$g_{cat}$ | Specific Yield of CNFs ($Y_C$), g/$g_{cat}$ |
| HMTA | 0.70 | | | 12.7 | 9.5 | 69.1 |
| Glycine | 0.33 | | | 23.5 | 15.4 | 154.7 |
| Urea | 0.27 | 723 | 1 | 31.7 | 11.3 | 100.9 |
| Starch | 0.41 | | | 31.7 | 12.8 | 95.7 |
| Citric acid | 0.26 | | | 26.2 | 17.1 | 171.3 |
| Oxalic acid | 0.06 | | | 20.4 | 11.9 | 120.9 |

The reaction of SCS with the catalyst takes place in three stages. The first is carried out at stoichiometric equilibrium yst = 0. The second case takes place when one has the lack of fuel in relation to stoichiometric equilibrium yst < 0. In some cases, there can be an excess of the concentration of fuel compared to stoichiometric equilibrium one yst > 0.

$$\varphi = y/y_{st}, \tag{1}$$

where y is the real molar ratio of fuel to catalyst and is the fuel excess. Fuel excess coefficient: the deviation of the mixture's fuel content from the stoichiometric amount (φ = 1). The value of fuel excess was set on the basis of reaching the constant weight of the mixture of

nickel and aluminum nitrates with fuel in order to keep the total weight of catalyst placed in the crucible constant before heating and SCS began. According to this, the weight of the initial mixture placed in a crucible before stating the SCS was the same. The fuel excess values for each fuel are shown in Table 1. The assumption of a constant weight of mixture was chosen according to preliminary experiments, which were taken at $\varphi = 1$ for each fuel. The majority of the time, the $Ni(NO_3)_2 \cdot 6H_2O\text{-}Al(NO_3)_3 \cdot 9H_2O$-fuel mixture experienced strong adiabatic heating, resulting in the explosive release of liquid and severe overheating of the mixture. Because of this effect, the results of catalysts were difficult to reproduce, and the weight of the $Ni(NO_3)_2 \cdot 6H_2O\text{-}Al(NO_3)_3 \cdot 9H_2O$-fuel mixture was assumed to be the same in all cases, with the only difference being the amount of the fuel.

The synthesized samples of catalysts were tested in a vertical catalytic quartz reactor, the schematic diagram of which is shown in Figure 2.

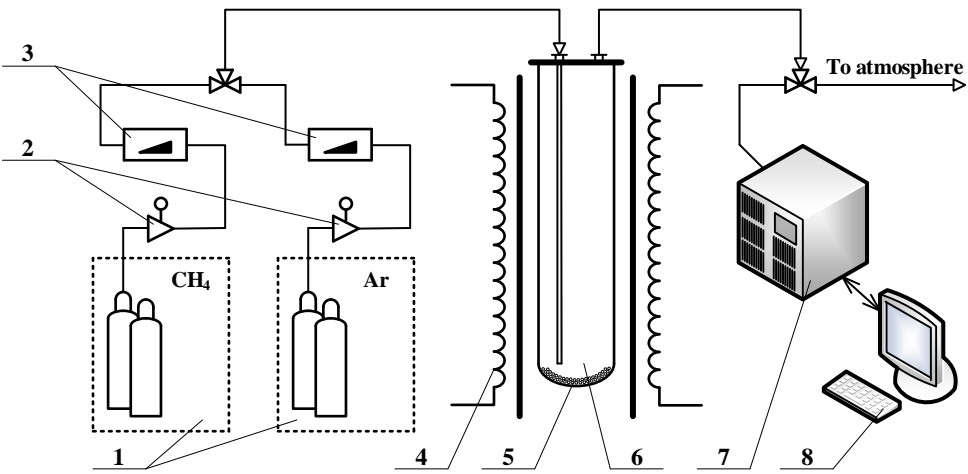

**Figure 2.** Scheme of catalytic setup for testing the decomposition of methane: 1—gas vessel; 2—reducing valve; 3—mass flow controller; 4—furnace; 5—catalyst; 6—vertical reactor; 7—gas chromatograph; 8—PC.

The reactor consists of the following units: gases ($CH_4$ and Ar) were fed into the flow through the quartz reactor (6) from cylinders (1) through pressure-reducing valves (2). Argon was fed only at the beginning of furnace heating; after reaching the temperature of synthesis (550 °C), pure methane was fed into the reactor and the argon was switched off. The gas flow rate was maintained at a given level by automatic mass flow controllers (3). The unreduced catalyst (5), with a weight of 12 mg, was loaded inside the reactor. The reactor was heated to a temperature of 550 °C in a resistance furnace (4). The synthesis was carried out at atmospheric pressure (1 atm). The catalyst was heated to its operating temperature in an inert gas flow (Ar), upon reaching which the gas was replaced by methane ($CH_4$). A methane–hydrogen mixture at the outlet of the reactor was taken for analysis in continuous mode. The passing of reaction was analyzed using a Khromos GC-1000-7 (Dzerzhinsk, Russia) gas chromatograph. The specific yield of carbon and hydrogen was calculated using the mass balance and data from gas chromatography.

$$x_{CH4} = (C_{H2}/(200 - C_{H2})) \cdot 100, \tag{2}$$

$$Y_C = (m_C - m_{cat})/m_{cat}, \tag{3}$$

$$Y_{H2} = (G_{CH4} \cdot x_{CH4} \cdot t)/(V \cdot 100), \tag{4}$$

where $x_{CH4}$ is the conversion of methane, vol. %; $C_{H2}$ is the concentration of hydrogen, vol.%; $Y_C$ is the specific yield of carbon, $g/g_{cat}$; $m_C$ is the weight of carbon formed, g; $m_{cat}$ is the weight of catalyst, g; $Y_{H2}$ is specific yield of hydrogen, $mol/g_{cat}$; $G_{CH4} = 100$ L/(h·$g_{cat}$); $x_{CH4}$ is the average conversion of methane, vol.%; t is the time until reaching full deactivation of the catalyst, h; V = 22.4 is the molar volume of gas, mol/L.

The phase composition of the synthesized samples was studied using X-ray diffraction (XPD) on a DRON-3 diffractometer (CuKα radiation, λ = 0.15406 nm). The crystallite size was calculated using Scherrer equation:

$$L = (k \cdot \lambda)/(B \cdot \cos\theta), \tag{5}$$

where L is the size of crystallite, Å; k = 0.9 is a constant value; B is FWHM, radians; θ is the diffraction angle, °.

HTREM (high resolution transmission electron microscopy) images were obtained using JEM-2200FS transmission electron microscope (JEOL Ltd., Tokyo, Japan) equipped with a Cs-corrector.

Low-temperature nitrogen adsorption (Quantachrome NOVA 1000e, Boynton Beach, FL, USA) was used to study the catalyst and carbon nanofibers that were made. The samples were preliminarily degassed at 100 °C for 3 h in order to remove the adsorbed gases and water. Full adsorption and desorption isotherms were plotted at 77 K and relative pressures $P/P_0$ 0.005–0.995. The specific surface area was calculated according to the Brunauer–Emmett–Teller (BET) method. The Barrett–Joyner–Hallenda (BJH) method was used to obtain the pore size distribution.

## 3. Results

The summary of the results obtained during the decomposition of methane over catalysts prepared by SCS using various fuels is presented in Table 1.

All the samples tested were not reduced. The majority of the data reported on methane decomposition was obtained using catalysts after hydrogen reduction [4,27,35,36]. The advantage of the catalysts under investigation, obtained by solution combustion synthesis, is that they need no reduction [15,37]. This has a positive effect on the duration of the chemical process by shortening its operation and making it more economically attractive for scale-up. A gradual reduction of catalyst occurred due to the formation of hydrogen as a result of the reaction. The maximum activity was shown by unreduced catalysts reaching the $H_2$ concentration at a level of 17–33%. The highest yield of hydrogen was achieved for an SCS catalyst prepared using citric acid (17.1 mol/$g_{cat}$). Figure 3a shows the dynamics of changes in the hydrogen concentration over the period until reaching the deactivation of the synthesized catalyst samples. The conversion of methane is shown in Figure 3b. The majority of catalysts converted at a rate of 17–19%. Catalysts synthesized using starch and urea as fuels possessed initial conversion levels of 14.5% and 9.4%, respectively. Their lifetime, however, was longer when compared to other fuel-based solution combustion synthesized samples. The delayed formation of carbon on the surface of the catalyst induces the extension of the time of operation (i.e., lifetime) of the latter during methane decomposition.

This catalyst had a relatively short total reaction time until deactivation, but the concentration of hydrogen formed was 28–29% for the first 16 h, then dropped. The initial conversion of methane possessed a lower value than is typical for a single-component nickel catalyst. Similar values were reached after 100–150 min of hydrogen and carbon synthesis over 50% Ni/$Al_2O_3$ and 50% Ni–5% Cu/$Al_2O_3$, which were prepared using the impregnation method [38]. The higher concentration of hydrogen was achieved for the glycine-based catalyst (30% for 8 h), after which it started to decrease. The longest time of catalyst operation was detected for starch and urea, but initially the hydrogen concentration was low compared to other fuels, such as HMTA, glycine, citric acid, and oxalic acid. It is worth noting that four fuels (i.e., HMTA, glycine, citric acid, and oxalic acid) possessed almost the same initial concentration of hydrogen formed during the first hour of reaction. The hydrogen concentration gradually decreases until the catalyst is saturated with solid reaction products (namely CNFs), as a result of which a sharp decrease in catalytic activity occurs within 2–3 h, leading to its deactivation. XRD patterns of catalysts are shown in Figure 4.

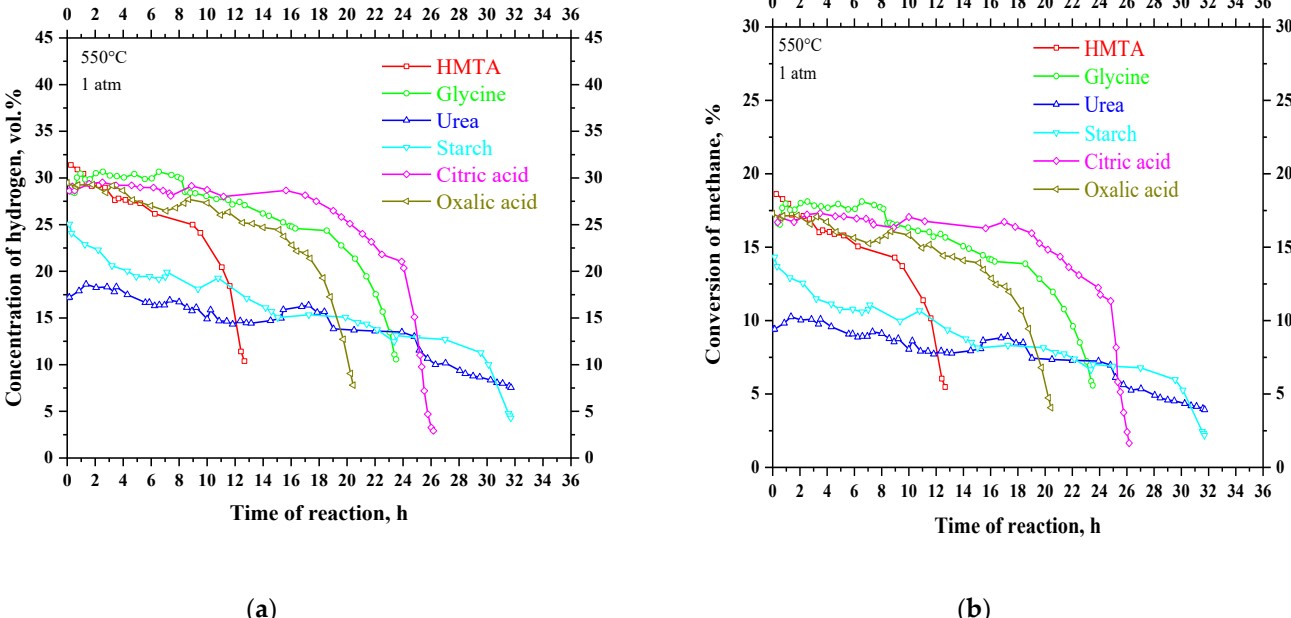

(**a**) (**b**)

**Figure 3.** Hydrogen concentration (**a**) and conversion of methane (**b**) vs. time of reaction for catalysts synthesized by SCS using various fuels (450 °C, 1 atm).

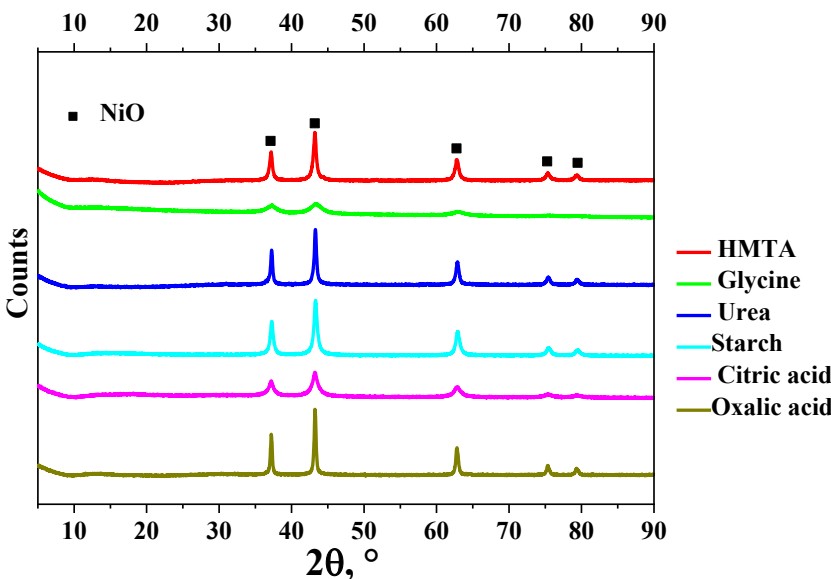

**Figure 4.** XRD patterns of catalysts synthesized.

After solution combustion synthesis, the NiO phase (JCPDS no. 00-047-1049) formed, which was shown by the position and intensity of the diffraction peaks for the catalysts. There was a significant problem during the thermodynamic calculations and estimation of fuel excess: the degree of polymerization of starch, since it is a polymer. The additional research on starch application in the solution combustion synthesis of nickel-based catalysts will be presented in the next paper.

Figure 5 shows that XRD patterns of CNFs made over SCS catalysts only showed the carbon phase (JCPDS no. 01-075-1621). Since there are not many nickel nanoparticles in the sample, there was no Ni phase in the XRD patterns of the CNFs that were synthesized. The intense (001) basal plane reflection in CNFs is clearly seen in diffraction patterns (Figure 5).

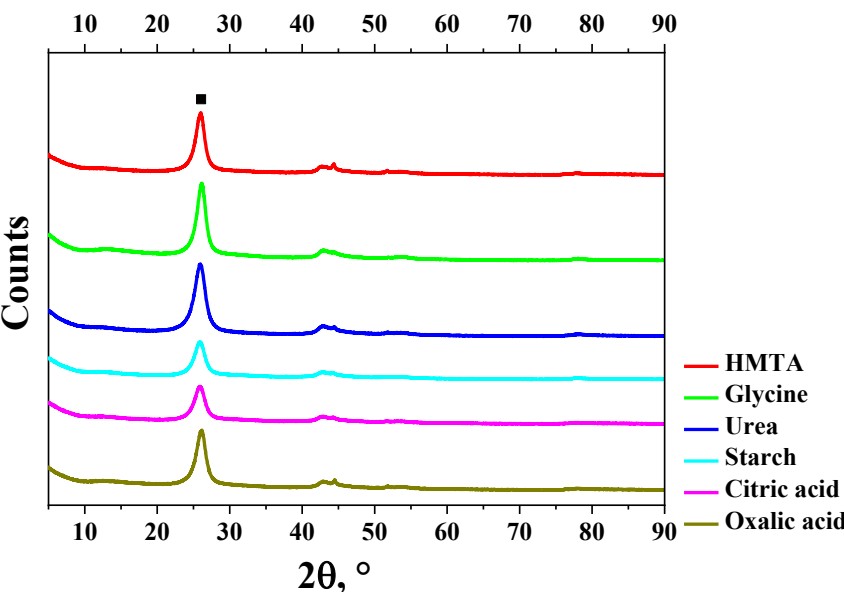

**Figure 5.** XRD patterns of carbon nanomaterials formed over solution combustion synthesized catalysts (450 °C, 1 atm) ((002) reflection of carbon phase is shown in the figure).

Table 2 shows summary of low temperature nitrogen adsorption data and XRD data.

**Table 2.** Summary data on the properties of catalysts and CNFs.

| Sample | Fuel Excess (φ) | Properties of Catalysts * | | | | Properties of CNFs | | | |
|---|---|---|---|---|---|---|---|---|---|
| | | XRD | BET | | | XRD | BET | | |
| | | $L_{av}$ (NiO), nm | $S_{sp}$, m²/g | $V_{total}$, cm³/g | $L_{pore}$, nm | $L_{av}$, nm | $S_{sp}$, m²/g | $V_{total}$, cm³/g | $L_{pore}$, nm |
| HMTA | 0.70 | 39.5 | 11 | 0.05 | 19.6 | 39.4 | 51 | 0.12 | 9.7 |
| Glycine | 0.33 | 16.6 | 153 | 0.50 | 13.1 | 21.4 | 61 | 0.24 | 15.5 |
| Urea | 0.27 | 37.2 | 51 | 0.21 | 16.7 | 22.9 | 86 | 0.16 | 7.8 |
| Starch | 0.41 | 35.8 | 81 | 0.27 | 13.5 | 17.7 | 86 | 0.17 | 8.0 |
| Citric acid | 0.26 | 20.9 | 94 | 0.23 | 9.9 | 21.9 | 80 | 0.16 | 7.9 |
| Oxalic acid | 0.06 | 40.2 | 49 | 0.30 | 24.4 | 25.9 | 74 | 0.14 | 7.6 |

* $L_{av}$—average crystallite size (according to XRD). $S_{sp}$—specific surface area. $V_{total}$—total pore volume. $L_{pore}$—average pore diameter.

It has been experimentally established that the type of fuel affects the characteristics of synthesized catalyst samples, so the surface area of the catalysts increases in the range of 49–153 m²/g for different fuels. According to the error of BET surface area measurements (the error of the analyzer was 5%), the surface area of all catalysts differs significantly when compared to each other. In some cases, a low surface area (HMTA-based catalyst, 11 m²/g) was obtained since the temperature of combustion was high enough to lead to the enlargement of nanoparticles. This is also caused by higher fuel excess coefficients, among other factors that were investigated.

The highest surface area was obtained for glycine fuel. The data on surface area is in agreement with XRD data. This sample showed the lowest average crystallite size (16.6 nm) among all catalysts. One of the largest crystallites and the lowest surface area were detected for the HMTA-based catalyst. The relatively high temperature during solution combustion, which causes the coarsening of nanoparticles, is most likely to blame for this. All catalysts were fully mesoporous. There was only one exclusion: the catalyst prepared using glycine, which possessed 5.8% of micropores (153 m²/g was the total surface area; 9 m²/g was the surface area of micropores). Two samples with high concentrations of hydrogen were glycine- and citric acid-based catalysts. These samples showed a relatively stable concentration of hydrogen over time and a relatively long lifetime of the catalyst.

They possessed relatively high surface areas (153 and 94 m$^2$/g, respectively), low crystallite sizes (16.6 and 20.9 nm, respectively), and low average pore diameters (13.1 nm and 9.9 nm, respectively). This factor has a direct effect on catalysis.

The surface area of CNFs synthesized over SCS catalysts was in the range of 51–86 m$^2$/g. It is difficult to find a clear relationship between the surface area of the catalyst and the CNFs formed. All CNFs contained micropores, excluding glycine, which consisted entirely of mesopores. It can be caused by full carbon formation and a longer lifetime of the catalyst during the reaction. The samples contained micropores as follows (the fraction of micropores in total surface area is presented in brackets): HMTA 5 m$^2$/g (9.8%); urea 21 m$^2$/g (24.4%); starch 20 m$^2$/g (23%); citric acid 23 m$^2$/g (28.75%); oxalic acid 12 m$^2$/g (16%).

Figure 6 shows typical TEM images of the SCS catalysts. The samples were represented by catalytic nanoparticles forming dense aggregates.

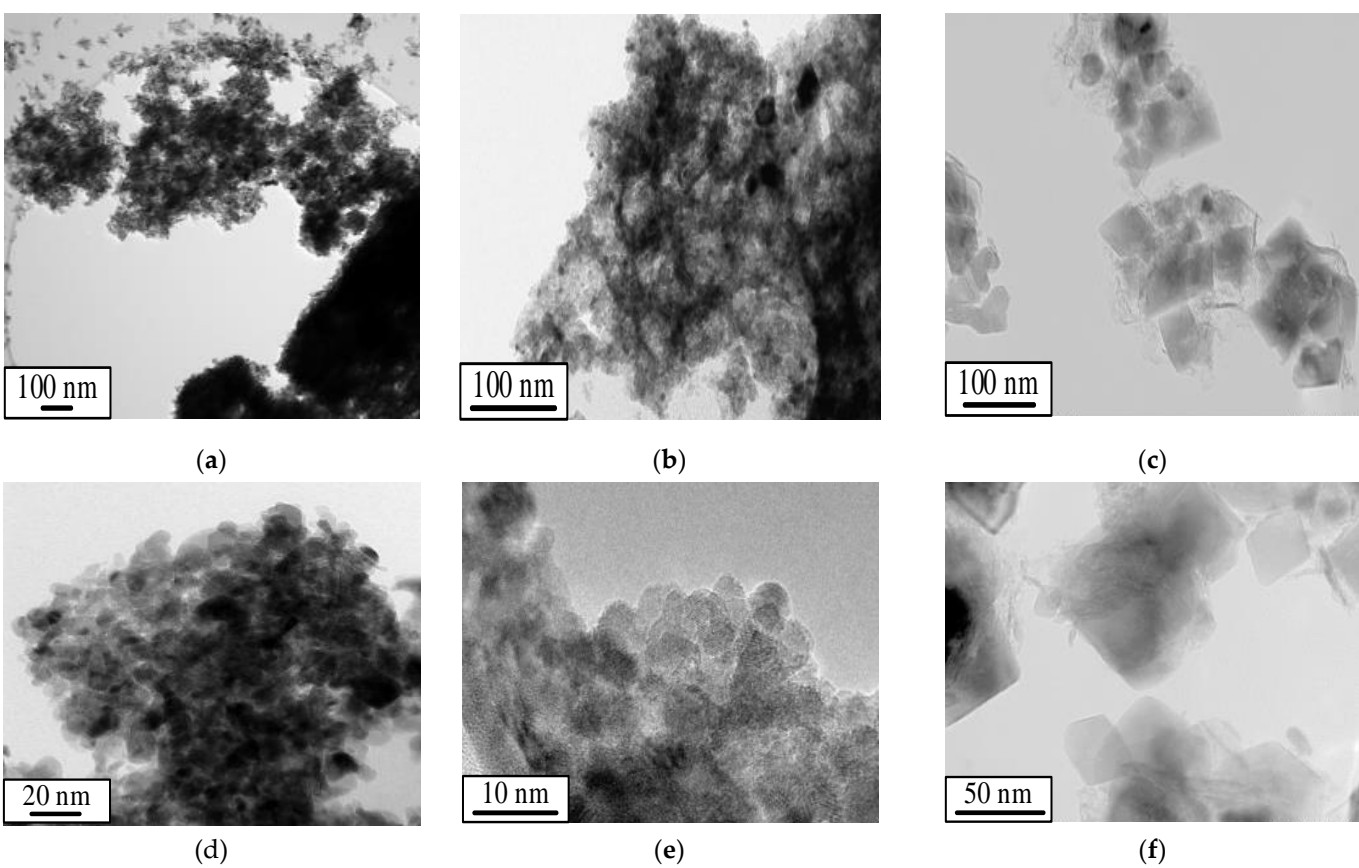

**Figure 6.** Typical TEM images of catalysts synthesized using solution combustion and various fuels: (**a**,**d**) HMTA; (**b**,**e**) glycine; (**c**,**f**) oxalic acid.

The synthesized catalyst samples possessed a homogeneous structure of catalytic nanoparticles, in which the NiO phase is uniformly distributed. Al$_2$O$_3$ can be considered as a promoting additive, not as substrate, since its amount is low enough compared to Ni. Usually, the quantity of Al$_2$O$_3$ is higher compared to content of active component (Ni or Ni-Cu) [38,39], but in our case the SCS was directed to synthesis of catalyst with high concentration of nickel. According to elemental analysis, the catalyst mainly contains nickel and oxygen. According to TEM data (Figure 6a–c), the size of the catalyst particles ranged from 10 to 20 nm. The crystallite size of the NiO phase in catalysts based on oxalic acid fuel showed the highest value compared to other fuels, which is in agreement with TEM images (Figure 6c). The crystallite size of this catalyst was close to values obtained for HMTA-based catalysts. This effect can be caused by high temperatures as a result of solution combustion

in the case of the application of high-energy fuels. This causes nanoparticles to grow in size, as well as crystallite sizes to grow in size.

Figure 7 shows TEM images of CNFs formed over solution combustion-based catalysts.

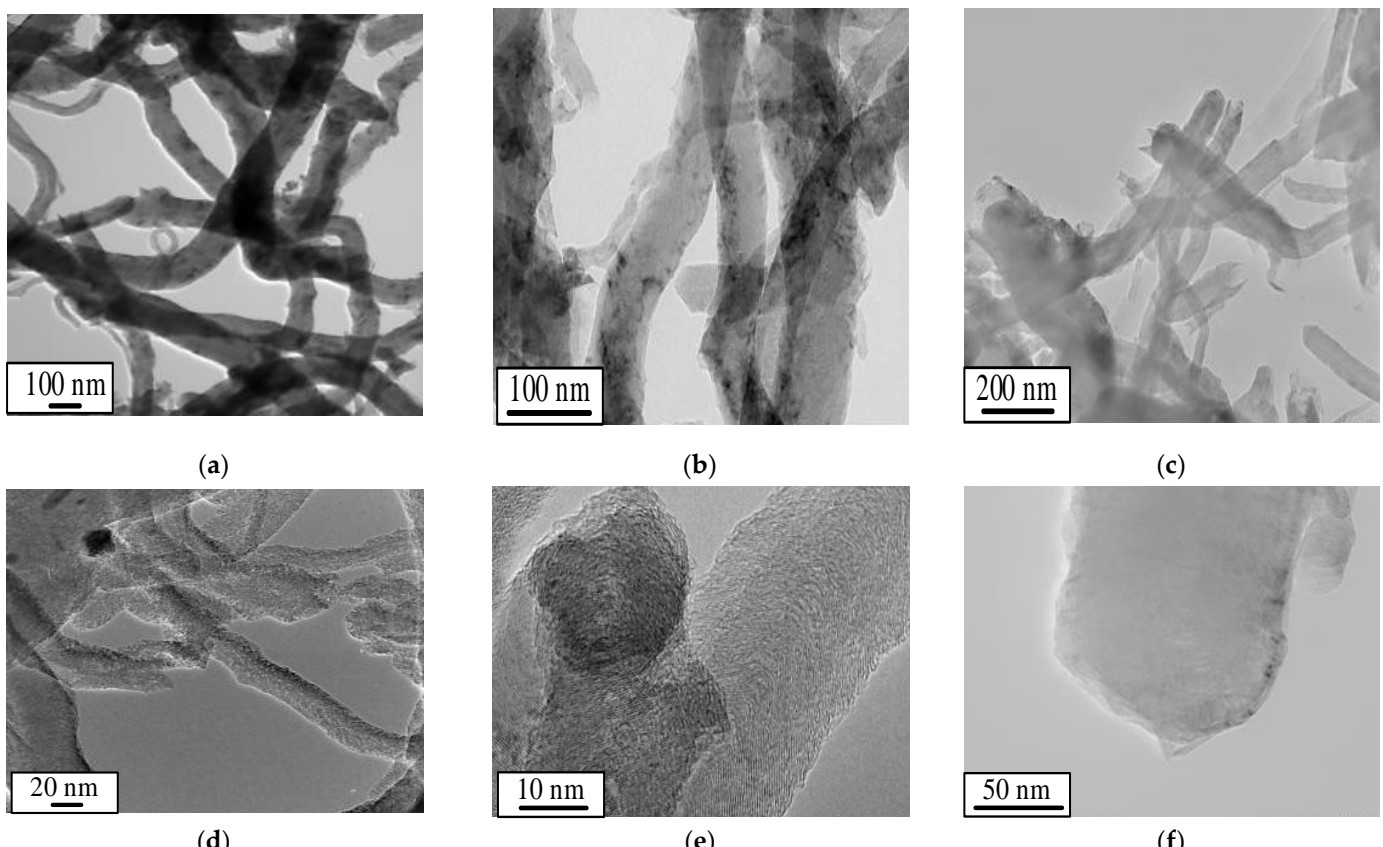

**Figure 7.** Typical TEM images of CNFs formed over solution combustion-based catalysts and various fuels: (**a**,**d**) HMTA; (**b**,**e**) glycine; (**c**,**f**) oxalic acid.

The carbon nanomaterial that was made had a porous structure made of tightly woven nanofibers that were between 100 and 150 nm in diameter. Micrographs showed that the sample also contains a small amount of catalyst nanoparticles, which are the active sites of the catalyst. CNFs had an almost "fishbone" structure [40]. The surface of the CNFs had a significant roughness. Because the reaction was continued until the catalyst was completely deactivated, the Ni nanoparticles were completely covered with a carbon shell (Figure 7d). Typically, for materials based on Ni-containing catalysts for methane decomposition and operating until complete deactivation, the metal nanoparticles are completely covered with carbon shells, and there are no carbon-free places on the surface of nanoparticles (as confirmed by X-ray photoelectron spectroscopy in [9] for similar material).

The results of experiments using different types of reducing agents were compared to data that had already been published (Table 3). Compared to other ways of making catalysts, the samples that were made by this method performed better in terms of surface area, pore volume, and average pore size. As one can see, combustion is a more efficient way to make the nickel catalyst on an alumina substrate with small crystallites and a higher specific area than the impregnation method. The latter is a very frequently used method for the preparation of nickel-containing catalytic systems on $Al_2O_3$ or $SiO_2$ substrates. Moreover, all the articles mentioned in Table 3 were applied to the two main steps of catalyst preparation. To begin, the alumina was primarily prepared using the sol-gel method. Then, these nickel-based catalysts were prepared via the wet impregnation method. These facts support the main advantage of SCS-based catalysts being the one-step preparation approach.

**Table 3.** A comparison of data on texture properties of catalysts synthesized using various methods.

| Catalyst | Preparation Technique | $L_{av}(NiO)$, nm | $S_{sp}$, m$^2$/g | $V_{total}$, cm$^3$/г | $L_{pore}$, nm | Ref. |
|---|---|---|---|---|---|---|
| 90Ni/10Al$_2$O$_3$ | solution combustion synthesis (citric acid fuel) | 20.9 | 94 | 0.23 | 9.97 | This work |
| 90Ni/10Al$_2$O$_3$ | solution combustion synthesis (glycine fuel) | 16.6 | 153 | 0.50 | 13.12 | |
| 90Ni/10Al$_2$O$_3$ | solution combustion synthesis (HMTA fuel) | 46.3 | 107 | 0.25 | 20.96 | [15] |
| 90Ni/SiO$_2$ | impregnation | 40 | – | – | – | [41] |
| 60Ni-10Al$_2$O$_3$ | impregnation | 26.6 | 66.1 | 0.13 | 7.2 | [42] |
| 60Ni-10Al$_2$O$_3$ | impregnation | 36.9 | 58.7 | 0.67 | 3.92 | [43] |
| 50Ni/Al$_2$O$_3$ | impregnation | 24.5 | 89 | 0.2 | 7.93 | [38] |

Table 4 shows a comparison of data regarding how effective catalysts are at making carbon and hydrogen. The data on conversion and yield of carbon formed were presented for a relatively low temperature (550 °C); therefore, the conversion of hydrogen was lower compared to some articles [6,38,43]. Usually, the research on methane decomposition is carried out in a mixture of CH$_4$ and N$_2$. In this work, we used pure methane to somehow reach the real conditions of catalyst operation on a laboratory scale. The significant drawback of the data reported for catalysts obtained using different techniques is the low time of the process carried out, e.g., 360 min [6], 180 min [8], and 650 min [38]. In some papers, the yield of carbon was low enough and in some cases did not reach at least 10 g/g$_{cat}$ [8], which is also caused by low time of reaction.

**Table 4.** A comparison of data reported for various catalysts used for methane decomposition and their efficiency in terms of formation of carbon and hydrogen.

| Ref. | Catalyst | Preparation Technique | Inlet Gas | Parameters of Process | Conversion of Hydrogen x, % Initial/Maximum | Yield of Carbon Yc, g/g$_{cat}$ |
|---|---|---|---|---|---|---|
| [6] | Ni/SiO$_2$ | wet impregnation | (1:4) CH$_4$/N$_2$ | 550 °C | 19/28 | - |
| [38] | Ni-Cu/Al$_2$O$_3$ | wet impregnation | (3:7) CH$_4$/N$_2$ | 750 °C | 84/85 | - |
| [42] | 50% Ni/Al$_2$O$_3$ | wet impregnation | (3:7) CH$_4$/N$_2$ | 625 °C, 1 bar | n/a/54 | - |
| [15] | 90%Ni/Al$_2$O$_3$ | solution combustion synthesis | Pure CH$_4$ | 535 °C, 1 bar | n/a | 268.3 |
| [44] | 3%Ni-15%Fe/MgO | co-precipitation | (1.5:1) CH$_4$/N$_2$ | 700 °C, | 64/73 | - |
| This work | 90%Ni/Al$_2$O$_3$ | solution combustion synthesis | Pure CH$_4$ | 550 °C, 1 bar | 16/18 | 171.3 |

## 4. Conclusions

The paper discusses high-percentage 90% Ni/Al$_2$O$_3$ catalysts that were made by burning different fuels in a solution. It has been established that the type of fuel affects the catalytic activity significantly. According to the results of the catalytic reaction, citric acid possessed the maximum specific yield of hydrogen and carbon nanofibers. The longer time of reaction was shown by starch and urea as fuels, despite their low methane conversion. The highest surface area of SCS-based catalysts was achieved in the case of glycine as fuel (153 m$^2$/g), which corresponded to the lowest pore diameter among other catalysts. The highest hydrogen and carbon nanofiber yields were obtained for catalysts with the use of citric acid (17.1 mol/g$_{cat}$, 171.3 g/g$_{cat}$).

**Author Contributions:** Conceptualization, P.B.K. and A.G.B.; methodology, A.M.; investigation, E.A.M., A.V.I. and A.V.U. All authors have read and agreed to the published version of the manuscript.

**Funding:** This work was financially supported by the State Task of Ministry of Science and Higher Education of Russia (FSUN-2023-0008).

**Institutional Review Board Statement:** Not applicable.

**Informed Consent Statement:** Not applicable.

**Data Availability Statement:** Data available from corresponding author upon a reasonable request.

**Conflicts of Interest:** The authors declare no conflict of interest.

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
