# Peer review of "Solution Combustion Synthesis of Ni/Al2O3 Catalyst for Methane Decomposition: Effect of Fuel"

_applsci, doi:10.3390/app13063962_

Round 1

Reviewer 1 Report

The work is interesting and well supported by adequately selected research techniques (XRD, HTREM, S BET). However, there is a lack of studies on catalyst activity over time and its regeneration.  Activity certainly decreases due to the formation of carbonaceous material (it can be clearly observed in Figure 3). It would be good to test the possibility of regeneration of the catalyst and its activity after regeneration. This could also be evaluated using XPS tests.

Author Response

Dear Reviewer,

Thank you for your valuable remarks. our answer is below and all changes in our paper were kept tracked.

The investigation of catalyst activity was carried out. Figures 3a and 3b indicate the operation of catalyst in time until reaching the deactivation. Regeneration of these catalysts was not carried out. out, since one should get two products simultaneously, CNFs and hydrogen. For such catalysts, on which the CNFs were grown, the regeneration could not be carried out because it meant the
complete burning out of CNFs The latter is an important product in terms of its application, i.e., the sale of which increases the economic efficiency of technology. The second drawback of regeneration is the formation of COx during the burning out of carbon that led to non-ecological
friendly process again. Concerning the XPS test: The XPS tests of CNFs with the catalyst inside give only the content of carbon and oxygen as elements in CNFs. Since the catalytic nanoparticles are covered with a carbon shell, there is no Ni or any other element signal detected using XPS.

Reviewer 2 Report

The paper presents high-percentage 90% Ni/10% Al2O3 catalysts, which were synthesized by solution combustion synthesis using various fuels. The authors have been established that the type of fuel affects the catalytic activity significantly. The research topic fits the Applied Sciences. Research topic is interesting for scientists, engineers and entrepreneurs. However, quality of this paper should be improved.

I have the following comments:

1.      Lines 19 and 101: the spaces should be removed, i.e. instead of “450 °C” should be “450°C” and instead of “1 °C” should be “1°C”.

2.      Lines 85, 143, 146 and 309: the space should be removed, i.e. instead of “550 °C” should be “550°C”.

3.      Lines 93, 128 and 131: Expressions are missing subscripts: “Ni(NO3)26H2O and Al(NO3)39H2O”.

4.      Line 168: instead of “100 °C” should be “100°C”.

5.      Line 202: Expression is missing subscripts: “Cu/Al2O3”.

6.      Line 217: Expression is missing subscripts: “C2H3NOn”.

7.      Line 320: Expression is missing subscripts: “Al2O3”.

8.      In the Abstract and Summary, different provisions regarding the composition of the catalyst are used (in the Abstract: “90% Ni/Al2O3”, in the Conclusions: “90% Ni/10% Al2O3”).  

9.      No error analysis.  

Author Response

Dear Reviewer,

Thank You so much for your great input. Our answers are given below.

1. Lines 19 and 101: the spaces should be removed, i.e. instead of “450 °C” should be
“450°C” and instead of “1 °C” should be “1°C”.
Answer:
The spaces have been removed.

2. Lines 85, 143, 146 and 309: the space should be removed, i.e. instead of “550 °C” should
be “550°C”.
Answer:
The spaces have been removed.

3. Lines 93, 128 and 131: Expressions are missing subscripts: “Ni(NO3)26H2O and
Al(NO3)39H2O”.
Answer:
The subscripts were corrected.
4.      Line 168: instead of “100 °C” should be “100°C”.
Answer:
The space was removed.

5.      Line 202: Expression is missing subscripts: “Cu/Al2O3”.
Answer:

The subscripts were corrected.

6.      Line 217: Expression is missing subscripts: “C2H3NOn”.
Answer:
It was corrected.

7.      Line 320: Expression is missing subscripts: “Al2O3”.
Answer:
The subscripts were corrected.

8.      In the Abstract and Summary, different provisions regarding the composition of the catalyst are used (in the Abstract: “90% Ni/Al2O3”, in the Conclusions: “90% Ni/10% Al2O3”).

Answer:
It was corrected.

9.      No error analysis.
Answer:
The paper mainly deals with average values, therefore the error analysis was not carried out. The BET surface area was rounded to meaningful values (the error of BET measurements was 5%). The crystallite size was presented as average values also. The span of nanofibers diameter is very wide therefore the only range was chosen. The comments on errors of BET surface area determination were
added.

Reviewer 3 Report

Report on the manuscript applsci-2275886-peer-review-v1 entitled “Solution combustion synthesis of Ni/Al2O3 catalyst for methane decomposition: Effect of fuel”.

The submitted manuscript should be revised. The following points should be addressed:

1. The submitted manuscript should be revised to be free from editing or grammar errors. For example, “XRD spectra of catalysts synthesized” should be “XRD spectra of the synthesized catalysts.”

2. The authors should indicate the reason of small BET surface area values and should compared with NiO without fuel or reported NiO.

3. In Fig. 6A which showed TEM image of catalyst synthesized using solution combustion and HMTA:, there is an up normal 1D nanostructure and it is not explained.

4. Editing error: figure 6 title was repeated after fig. 6 and fig. 7. Please, fix it?

5. The role of Al2O3 should be discussed in detail and followed by the mechanism of using NiO/Al2O3 as a catalyst.

Author Response

Dear Reviewer,

Please find our answers below:

Report on the manuscript applsci-2275886-peer-review-v1 entitled “Solution combustion synthesis of Ni/Al2O3 catalyst for methane decomposition: Effect of fuel”.

The submitted manuscript should be revised. The following points should be addressed:

  1. The submitted manuscript should be revised to be free from editing or grammar errors. For example, “XRD spectra of catalysts synthesized” should be “XRD spectra of the synthesized catalysts.”

Answer:

Thank you for comment. The manuscript was checked for grammar errors.

  1. The authors should indicate the reason of small BET surface area values and should compared with NiO without fuel or reported NiO.

Answer:

Table 6 shows a comparison of BET surface area and Lav for Ni-based catalysts reported. The use of the parameters for preparation the catalyst without fuel did not lead to synthesis of samples with high surface area (below 10-20 m2/g). The solution combustion synthesis it is also impossible to be carried out without fuel.

The main reason of low specific surface area is high temperature occurred during combustion. For some fuels, it led to agglomeration of nanoparticles (e.g. HMTA) and small BET surface area.  

  1. In Fig. 6A which showed TEM image of catalyst synthesized using solution combustion and HMTA:, there is an up normal 1D nanostructure and it is not explained.

Answer:

Thank you for comment. Actually, the catalytic nanoparticles presented in TEM image in Fig. 6a are aggregated. But in TEM images with higher magnification it is seen that these are  nanoparticles (Fig. 6b). High temperature of formation (HMTA-based system possesses stronger adiabatic heating temperature) led to synthesis of such structures.

  1. Editing error: figure 6 title was repeated after fig. 6 and fig. 7. Please, fix it?

Answer:

Thank you for comment. It has been fixed.

  1. The role of Al2O3 should be discussed in detail and followed by the mechanism of using NiO/Al2O3 as a catalyst.

Answer:

The necessary comments were made. Alumina plays the role of promoting additive and its concentration higher compared to Ni. The catalyst is Ni/Al2O3, since oxide is reduced to metallic nickel during reaction.

“The synthesized catalyst samples possessed a homogeneous structure of catalytic nanoparticles, in which the NiO phase is uniformly distributed. Al2O3 can be considered as a promoting additive, not as substrate, since its amount is low enough compared to Ni. Usually, the quantity of Al2O3 is higher compared to content of active component (Ni or Ni-Cu) [38,39], but in our case the SCS was directed to synthesis of catalyst with high concentration of nickel.”

Round 2

Reviewer 3 Report

The submitted revision Could be accepted